# A Framework for Testing Identifiability of Bayesian Models of Perception

**Luigi Acerbi**[1,2]      **Wei Ji Ma**[2]      **Sethu Vijayakumar**[1]

[1] School of Informatics, University of Edinburgh, UK
[2] Center for Neural Science & Department of Psychology, New York University, USA
{luigi.acerbi,weijima}@nyu.edu    sethu.vijayakumar@ed.ac.uk

## Abstract

Bayesian observer models are very effective in describing human performance in perceptual tasks, so much so that they are trusted to faithfully recover hidden mental representations of priors, likelihoods, or loss functions from the data. However, the intrinsic degeneracy of the Bayesian framework, as multiple combinations of elements can yield empirically indistinguishable results, prompts the question of model identifiability. We propose a novel framework for a systematic testing of the identifiability of a significant class of Bayesian observer models, with practical applications for improving experimental design. We examine the theoretical identifiability of the inferred internal representations in two case studies. First, we show which experimental designs work better to remove the underlying degeneracy in a time interval estimation task. Second, we find that the reconstructed representations in a speed perception task under a slow-speed prior are fairly robust.

## 1   Motivation

Bayesian Decision Theory (BDT) has been traditionally used as a benchmark of ideal perceptual performance [1], and a large body of work has established that humans behave close to Bayesian observers in a variety of psychophysical tasks (see e.g. [2, 3, 4]). The efficacy of the Bayesian framework in explaining a huge set of diverse behavioral data suggests a stronger interpretation of BDT as a *process model* of perception, according to which the formal elements of the decision process (priors, likelihoods, loss functions) are independently represented in the brain and shared across tasks [5, 6]. Importantly, such mental representations, albeit not directly accessible to the experimenter, can be tentatively recovered from the behavioral data by 'inverting' a model of the decision process (e.g., priors [7, 8, 9, 10, 11, 12, 13, 14], likelihood [9], and loss functions [12, 15]). The ability to faithfully reconstruct the observer's internal representations is key to the understanding of several outstanding issues, such as the complexity of statistical learning [11, 12, 16], the nature of mental categories [10, 13], and linking behavioral to neural representations of uncertainty [4, 6].

In spite of these successes, the validity of the conclusions reached by fitting Bayesian observer models to the data can be questioned [17, 18]. A major issue is that the inverse mapping from observed behavior to elements of the decision process is not unique [19]. To see this degeneracy, consider a simple perceptual task in which the observer is exposed to stimulus $s$ that induces a noisy sensory measurement $x$. The Bayesian observer reports the optimal estimate $s^*$ that minimizes his or her expected loss, where the loss function $\mathcal{L}(s, \hat{s})$ encodes the loss (or cost) for choosing $\hat{s}$ when the real stimulus is $s$. The optimal estimate for a given measurement $x$ is computed as follows [20]:

$$s^*(x) = \arg\min_{\hat{s}} \int q_{\text{meas}}(x|s) q_{\text{prior}}(s) \mathcal{L}(s, \hat{s}) \, ds \tag{1}$$

where $q_{\text{prior}}(s)$ is the observer's prior density over stimuli and $q_{\text{meas}}(x|s)$ the observer's sensory likelihood (as a function of $s$). Crucially, for a given $x$, the solution of Eq. 1 is the same for any

triplet of prior $q_{\text{prior}}(s) \cdot \phi_1(s)$, likelihood $q_{\text{meas}}(x|s) \cdot \phi_2(s)$, and loss function $\mathcal{L}(\hat{s}, s) \cdot \phi_3(s)$, where the $\phi_i(s)$ are three generic functions such that $\prod_{i=1}^{3} \phi_i(s) = c$, for a constant $c > 0$. This analysis shows that the 'inverse problem' is ill-posed, as multiple combinations of priors, likelihoods and loss functions yield identical behavior [19], even before considering other confounding issues, such as latent states. If uncontrolled, this redundancy of solutions may condemn the Bayesian models of perception to a severe form of *model non-identifiability* that prevents the reliable recovery of model components, and in particular the sought-after internal representations, from the data.

In practice, the degeneracy of Eq. 1 can be prevented by enforcing constraints on the shape that the internal representations are allowed to take. Such constraints include: (a) theoretical considerations (e.g., that the likelihood emerges from a specific noise model [21]); (b) assumptions related to the experimental layout (e.g., that the observer will adopt the loss function imposed by the reward system of the task [3]); (c) additional measurements obtained either in independent experiments or in distinct conditions of the same experiment (e.g., through Bayesian transfer [5]). Crucially, both (b) and (c) are under partial control of the experimenter, as they depend on the experimental design (e.g., choice of reward system, number of conditions, separate control experiments). Although several approaches have been used or proposed to suppress the degeneracy of Bayesian models of perception [12, 19], there has been no systematic analysis – neither empirical nor theoretical – of their effectiveness, nor a framework to perform such study *a priori*, before running an experiment.

This paper aims to fill this gap for a large class of psychophysical tasks. Similar issues of model non-identifiability are not new to psychology [22], and generic techniques of analysis have been proposed (e.g., [23]). Here we present an efficient method that exploits the common structure shared by many Bayesian models of sensory estimation. First, we provide a general framework that allows a modeller to perform a systematic, a priori investigation of identifiability, that is the ability to reliably recover the parameters of interest, for a chosen Bayesian observer model. Second, we show how, by comparing identifiability within distinct ideal experimental setups, our framework can be used to improve experimental design. In Section 2 we introduce a novel class of observer models that is both flexible and efficient, key requirements for the subsequent analysis. In Section 3 we describe a method to efficiently explore identifiability of a given observer model within our framework. In Section 4 we show an application of our technique to two well-known scenarios in time perception [24] and speed perception [9]. We conclude with a few remarks in Section 5.

## 2 Bayesian observer model

Here we introduce a continuous class of Bayesian observer models parametrized by vector $\boldsymbol{\theta}$. Each value of $\boldsymbol{\theta}$ corresponds to a specific observer that can be used to model the psychophysical task of interest. The current model (class) extends previous work [12, 14] by encompassing any sensori-motor estimation task in which a one-dimensional stimulus magnitude variable $s$, such as duration, distance, speed, etc. is directly estimated by the observer. This is a fundamental experimental condition representative of several studies in the field (e.g., [7, 9, 12, 24, 14]). With minor modifications, the model can also cover angular variables such as orientation (for small errors) [8, 11] and multi-dimensional variables when symmetries make the actual inference space one-dimensional [25]. The main novel feature of the presented model is that it covers a large representational basis with a single parametrization, while still allowing fast computation of the observer's behavior, both necessary requirements to permit an exploration of the complex model space, as described in Section 3.

The generic observer model is constructed in four steps (Figure 1 a & b): 1) the *sensation stage* describes how the physical stimulus $s$ determines the internal measurement $x$; 2) the *perception stage* describes how the internal measurement $x$ is combined with the prior to yield a posterior distribution; 3) the *decision-making stage* describes how the posterior distribution and loss function guide the choice of an 'optimal' estimate $s^*$ (possibly corrupted by lapses); and finally 4) the *response stage* describes how the optimal estimate leads to the observed response $r$.

### 2.1 Sensation stage

For computational convenience, we assume that the stimulus $s \in \mathbb{R}^+$ (the *task space*) comes from a discrete experimental distribution of stimuli $s_i$ with frequencies $P_i$, with $P_i > 0, \sum_i P_i = 1$ for $1 \leq i \leq N_{\text{exp}}$. Discrete distributions of stimuli are common in psychophysics, and continu-

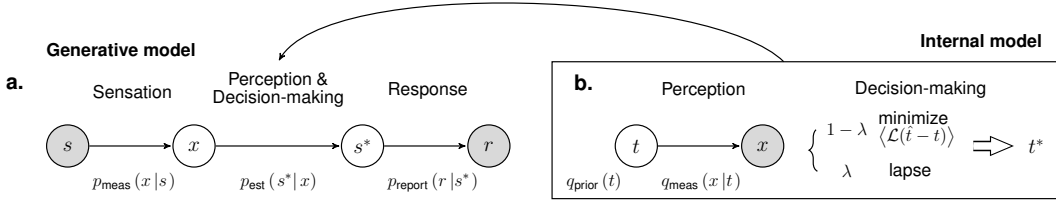

Figure 1: **Observer model.** Graphical model of a sensorimotor estimation task, as seen from the outside (**a**), and from the subjective point of view of the observer (**b**). **a: Objective generative model of the task.** Stimulus $s$ induces a noisy sensory measurement $x$ in the observer, who decides for estimate $s^*$ (see **b**). The recorded response $r$ is further perturbed by reporting noise. Shaded nodes denote experimentally accessible variables. **b: Observer's internal model of the task.** The observer performs inference in an internal measurement space in which the unknown stimulus is denoted by $t$ (with $t = f(s)$). The observer either chooses the subjectively optimal value of $t$, given internal measurement $x$, by minimizing the expected loss, or simply lapses with probability $\lambda$. The observer's chosen estimate $t^*$ is converted to task space through the inverse mapping $s^* = f^{-1}(t^*)$. The whole process in this panel is encoded in (**a**) by the estimate distribution $p_{\text{est}}(s^*|x)$.

ous distributions can be 'binned' and approximated up to the desired precision by increasing $N_{\text{exp}}$. Due to noise in the sensory systems, stimulus $s$ induces an internal measurement $x \in \mathbb{R}$ according to measurement distribution $p_{\text{meas}}(x|s)$ [20]. In general, the magnitude of sensory noise may be stimulus-dependent in task space, in which case the shape of the likelihood would change from point to point – which is unwieldy for subsequent computations. We want instead to find a transformed space in which the scale of the noise is stimulus-independent and the likelihood translationally invariant [9] (see Supplementary Material). We assume that such change of variables is performed by a function $f(s) : s \to t$ that monotonically maps stimulus $s$ from task space into $t = f(s)$, which lives with $x$ in an *internal measurement space*. We assume for $f(s)$ the following parametric form:

$$f(s) = A \ln\left[1 + \left(\frac{s}{s_0}\right)^d\right] + B \quad \text{with inverse} \quad f^{-1}(t) = s_0 \sqrt[d]{e^{\frac{t-B}{A}} - 1} \tag{2}$$

where $A$ and $B$ are chosen, without loss of generality, such that the discrete distribution of stimuli mapped in internal space, $\{f(s_i)\}$ for $1 \leq i \leq N_{\text{exp}}$, has range $[-1, 1]$. The parametric form of the sensory map in Eq. 2 can approximate both Weber-Fechner's law and Steven's law, for different values of base noise magnitude $s_0$ and power exponent $d$ (see Supplementary Material).

We determine the shape of $p_{\text{meas}}(x|s)$ with a maximum-entropy approach by fixing the first four moments of the distribution, and under the rather general assumptions that the sensory measurement is unimodal and centered on the stimulus in internal measurement space. For computational convenience, we express $p_{\text{meas}}(x|s)$ as a mixture of (two) Gaussians in internal measurement space:

$$p_{\text{meas}}(x|s) = \pi \mathcal{N}\left(x|f(s) + \mu_1, \sigma_1^2\right) + (1 - \pi)\mathcal{N}\left(x|f(s) + \mu_2, \sigma_2^2\right) \tag{3}$$

where $\mathcal{N}\left(x|\mu, \sigma^2\right)$ is a normal distribution with mean $\mu$ and variance $\sigma^2$ (in this paper we consider a two-component mixture but derivations easily generalize to more components). The parameters in Eq. 3 are partially determined by specifying the first four central moments: $\mathbb{E}[x] = f(s)$, $\text{Var}[x] = \sigma^2$, $\text{Skew}[x] = \gamma$, $\text{Kurt}[x] = \kappa$; where $\sigma, \gamma, \kappa$ are free parameters. The remaining degrees of freedom (one, for two Gaussians) are fixed by picking a distribution that satisfies unimodality and locally maximizes the differential entropy (see Supplementary Material). The sensation model represented by Eqs. 2 and 3 allows to express a large class of sensory models in the psychophysics literature, including for instance stimulus-dependent noise [9, 12, 24] and 'robust' mixture models [21, 26].

## 2.2 Perceptual stage

Without loss of generality, we represent the observer's prior distribution $q_{\text{prior}}(t)$ as a mixture of $M$ dense, regularly spaced Gaussian distributions in internal measurement space:

$$q_{\text{prior}}(t) = \sum_{m=1}^{M} w_m \mathcal{N}\left(t|\mu_{\min} + (m - 1)a, a^2\right) \qquad a \equiv \frac{\mu_{\max} - \mu_{\min}}{M - 1} \tag{4}$$

where $w_m$ are the mixing weights, $a$ the lattice spacing and $[\mu_{\min}, \mu_{\max}]$ the range in internal space over which the prior is defined (chosen $50\%$ wider than the true stimulus range). Eq. 4 allows the modeller to approximate any observer's prior, where $M$ regulates the fine-grainedness of the representation and is determined by computational constraints (for all our analyses we fix $M = 15$).

For simplicity, we assume that the observer's internal representation of the likelihood, $q_{\mathrm{meas}}(x|t)$, is expressed in the same measurement space and takes again the form of a unimodal mixture of two Gaussians, Eq. 3, although with possibly different variance, skewness and kurtosis (respectively, $\tilde{\sigma}^2$, $\tilde{\gamma}$ and $\tilde{\kappa}$) than the true likelihood. We write the observer's posterior distribution as: $q_{\mathrm{post}}(t|x) = \frac{1}{Z} q_{\mathrm{prior}}(t) q_{\mathrm{meas}}(x|t)$ with $Z$ the normalization constant.

## 2.3 Decision-making stage

According to Bayesian Decision Theory (BDT), the observer's 'optimal' estimate corresponds to the value of the stimulus that minimizes the expected loss, with respect to loss function $\mathcal{L}(t, \hat{t})$, where $t$ is the true value of the stimulus and $\hat{t}$ its estimate. In general the loss could depend on $t$ and $\hat{t}$ in different ways, but for now we assume a functional dependence only on the stimulus difference in internal measurement space, $\hat{t} - t$. The (subjectively) optimal estimate is:

$$t^*(x) = \arg\min_{\hat{t}} \int q_{\mathrm{post}}(t|x) \mathcal{L}\left(\hat{t} - t\right) dt \qquad (5)$$

where the integral on the r.h.s. represents the expected loss. We make the further assumption that the loss function is well-behaved, that is smooth, with a unique minimum at zero (i.e., the loss is minimal when the estimate matches the true stimulus), and with no other local minima. As before, we adopt a maximum-entropy approach and we restrict ourselves to the class of loss functions that can be described as mixtures of two (inverted) Gaussians:

$$\mathcal{L}(\hat{t} - t) = -\pi^\ell \mathcal{N}\left(\hat{t} - t | \mu_1^\ell, {\sigma_1^\ell}^2\right) - (1 - \pi^\ell)\mathcal{N}\left(\hat{t} - t | \mu_2^\ell, {\sigma_2^\ell}^2\right). \qquad (6)$$

Although the loss function is not a distribution, we find convenient to parametrize it in terms of statistics of a corresponding unimodal distribution obtained by flipping Eq. 6 upside down: Mode $[t'] = 0$, Var $[t'] = \sigma_\ell^2$, Skew $[t'] = \gamma_\ell$, Kurt $[t'] = \kappa_\ell$; with $t' \equiv \hat{t} - t$. Note that we fix the location of the mode of the mixture of Gaussians so that the global minimum of the loss is at zero. As before, the remaining free parameter is fixed by taking a local maximum-entropy solution. A single inverted Gaussian already allows to express a large variety of losses, from a delta function (MAP strategy) for $\sigma_\ell \to 0$ to a quadratic loss for $\sigma_\ell \to \infty$ (in practice, for $\sigma_\ell \gtrsim 1$), and it has been shown to capture human sensorimotor behavior quite well [15]. Eq. 6 further extends the range of describable losses to asymmetric and more or less peaked functions. Crucially, Eqs. 3, 4, 5 and 6 combined yield an analytical expression for the expected loss that is a mixture of Gaussians (see Supplementary Material) that allows for a fast numerical solution [14, 27].

We allow the possibility that the observer may occasionally deviate from BDT due to lapses with probability $\lambda \geq 0$. In the case of lapse, the observer's estimate $t^*$ is drawn randomly from the prior [11, 14]. The combined stochastic estimator with lapse in task space has distribution:

$$p_{\mathrm{est}}(s^*|x) = (1 - \lambda) \cdot \delta\left[s^* - f^{-1}\left(t^*(x)\right)\right] + \lambda \cdot q_{\mathrm{prior}}(s^*) |f'(s^*)| \qquad (7)$$

where $f'(s^*)$ is the derivative of the mapping in Eq. 2 (see Supplementary Material).

## 2.4 Response stage

We assume that the observer's response $r$ is equal to the observer's estimate corrupted by independent normal noise in task space, due to motor error and other residual sources of variability:

$$p_{\mathrm{report}}(r|s^*) = \mathcal{N}\left(r|s^*, \sigma_{\mathrm{report}}^2(s^*)\right) \qquad (8)$$

where we choose a simple parameteric form for the variance: $\sigma_{\mathrm{report}}^2(s) = \rho_0^2 + \rho_1^2 s^2$, that is the sum of two independent noise terms (constant noise plus some noise that grows with the magnitude of the stimulus). In our current analysis we are interested in observer models of perception, so we do not explicitly model details of the motor aspect of the task and we do not include the consequences of response error into the decision making part of the model (Eq. 5).

Finally, the main observable that the experimenter can measure is the response probability density, $p_{\text{resp}}(r|s; \boldsymbol{\theta})$, of a response $r$ for a given stimulus $s$ and observer's parameter vector $\boldsymbol{\theta}$ [12]:

$$p_{\text{resp}}(r|s; \boldsymbol{\theta}) = \int \mathcal{N}\left(r|s^*, \sigma_{\text{report}}^2(s^*)\right) p_{\text{est}}(s^*|x) p_{\text{meas}}(x|s) \, ds^* \, dx, \qquad (9)$$

obtained by marginalizing over unobserved variables (see Figure 1 a), and which we can compute through Eqs. 3–8. An observer model is fully characterized by parameter vector $\boldsymbol{\theta}$:

$$\boldsymbol{\theta} = \left(\sigma, \gamma, \kappa, s_0, d, \tilde{\sigma}, \tilde{\gamma}, \tilde{\kappa}, \sigma_\ell, \gamma_\ell, \kappa_\ell, \{w_m\}_{m=1}^M, \rho_0, \rho_1, \lambda\right). \qquad (10)$$

An *experimental design* is specified by a *reference observer model* $\boldsymbol{\theta}^*$, an experimental distribution of stimuli (a discrete set of $N_{\text{exp}}$ stimuli $s_i$, each with relative frequency $P_i$), and possibly a subset of parameters that are assumed to be equal to some a priori or experimentally measured values during the inference. For experiments with multiple conditions, an observer model typically shares several parameters across conditions. The reference observer $\boldsymbol{\theta}^*$ represents a 'typical' observer for the idealized task under examination; its parameters are determined from pilot experiments, the literature, or educated guesses. We are ready now to tackle the problem of identifiability of the parameters of $\boldsymbol{\theta}^*$ within our framework for a given experimental design.

## 3 Mapping a priori identifiability

Two observer models $\boldsymbol{\theta}$ and $\boldsymbol{\theta}^*$ are a priori *practically non-identifiable* if they produce similar response probability densities $p_{\text{resp}}(r|s_i; \boldsymbol{\theta})$ and $p_{\text{resp}}(r|s_i; \boldsymbol{\theta}^*)$ for all stimuli $s_i$ in the experiment. Specifically, we assume that data are generated by the reference observer $\boldsymbol{\theta}^*$ and we ask what is the chance that a randomly generated dataset $\mathcal{D}$ of a fixed size $N_{\text{tr}}$ will instead provide support for observer $\boldsymbol{\theta}$. For *one* specific dataset $\mathcal{D}$, a natural way to quantify support would be the posterior probability of a model given the data, $\Pr(\boldsymbol{\theta}|\mathcal{D})$. However, randomly generating a large number of datasets so as to approximate the expected value of $\Pr(\boldsymbol{\theta}|\mathcal{D})$ over *all* datasets, in the spirit of previous work on model identifiability [23], becomes intractable for complex models such as ours.

Instead, we define the *support* for observer model $\boldsymbol{\theta}$, given dataset $\mathcal{D}$, as its log likelihood, $\log \Pr(\mathcal{D}|\boldsymbol{\theta})$. The log (marginal) likelihood is a widespread measure of evidence in model comparison, from sampling algorithms to metrics such as AIC, BIC and DIC [28]. Since we know the generative model of the data, $\Pr(\mathcal{D}|\boldsymbol{\theta}^*)$, we can compute the expected support for model $\boldsymbol{\theta}$ as:

$$\langle \log \Pr(\mathcal{D}|\boldsymbol{\theta}) \rangle = \int_{|\mathcal{D}|=N_{\text{tr}}} \log \Pr(\mathcal{D}|\boldsymbol{\theta}) \Pr(\mathcal{D}|\boldsymbol{\theta}^*) \, d\mathcal{D}. \qquad (11)$$

The formal integration over all possible datasets with fixed number of trials $N_{\text{tr}}$ yields:

$$\langle \log \Pr(\mathcal{D}|\boldsymbol{\theta}) \rangle = -N_{\text{tr}} \sum_{i=1}^{N_{exp}} P_i \cdot D_{\text{KL}}\left(p_{\text{resp}}(r|s_i; \boldsymbol{\theta}^*)||p_{\text{resp}}(r|s_i; \boldsymbol{\theta})\right) + \text{const} \qquad (12)$$

where $D_{\text{KL}}(\cdot||\cdot)$ is the Kullback-Leibler (KL) divergence between two distributions, and the constant is an entropy term that does not affect our subsequent analysis, not depending on $\boldsymbol{\theta}$ (see Supplementary Material for the derivation). Crucially, $D_{\text{KL}}$ is non-negative, and zero only when the two distributions are identical. The asymmetry of the KL-divergence captures the different status of $\boldsymbol{\theta}^*$ and $\boldsymbol{\theta}$ (that is, we measure differences only on datasets generated by $\boldsymbol{\theta}^*$). Eq. 12 quantifies the average support for model $\boldsymbol{\theta}$ given true model $\boldsymbol{\theta}^*$, which we use as a proxy to assess model identifiability. As an empirical tool to explore the identifiability landscape, we define the *approximate expected posterior density* as:

$$\mathcal{E}\left(\boldsymbol{\theta}|\boldsymbol{\theta}^*\right) \propto e^{\langle \log \Pr(\mathcal{D}|\boldsymbol{\theta}) \rangle} \qquad (13)$$

and we sample from Eq. 13 via MCMC. Clearly, $\mathcal{E}(\boldsymbol{\theta}|\boldsymbol{\theta}^*)$ is maximal for $\boldsymbol{\theta} = \boldsymbol{\theta}^*$ and generally high for regions of the parameter space empirically close to the predictions of $\boldsymbol{\theta}^*$. Moreover, the peakedness of $\mathcal{E}(\boldsymbol{\theta}|\boldsymbol{\theta}^*)$ is modulated by the number of trials $N_{\text{tr}}$ (the more the trials, the more information to discriminate between models).

## 4 Results

We apply our framework to two case studies: the inference of priors in a time interval estimation task (see [24]) and the reconstruction of prior and noise characteristics in speed perception [9].

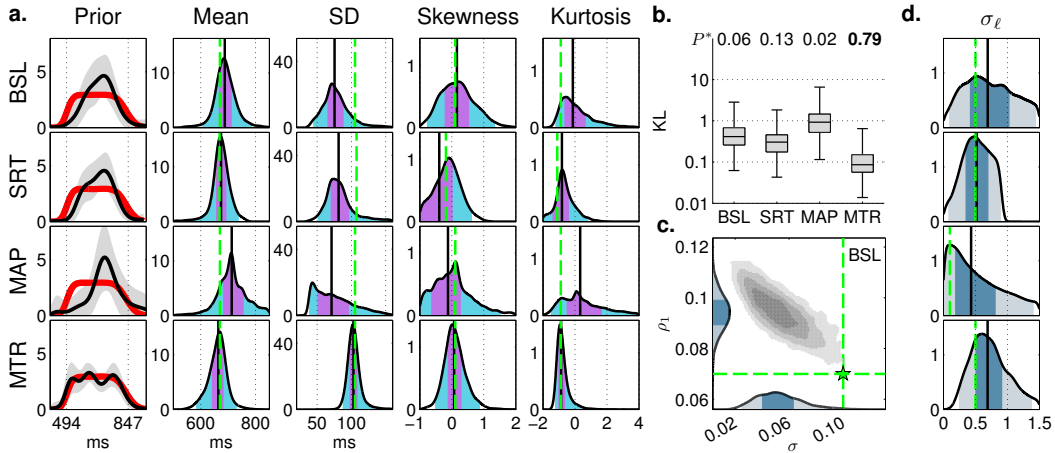

Figure 2: **Internal representations in interval timing (Short condition).** Accuracy of the reconstructed priors in the Short range; each row corresponds to a different experimental design. **a:** The first column shows the reference prior (thick red line) and the recovered mean prior $\pm$ 1 SD (black line and shaded area). The other columns display the distributions of the recovered central moments of the prior. Each panel shows the median (black line), the interquartile range (dark-shaded area) and the 95 % interval (light-shaded area). The green dashed line marks the true value. **b:** Box plots of the symmetric KL-divergence between the reconstructed priors and the prior of the reference observer. At top, the *primacy probability* $P^*$ of each setup having less reconstruction error than all the others (computed by bootstrap). **c:** Joint posterior density of sensory noise $\sigma$ and motor noise $\rho_1$ in setup BSL (gray contour plot; colored plots are marginal distributions). The parameters are anti-correlated, and discordant with the true value (star and dashed lines). **d:** Marginal posterior density for loss width parameter $\sigma_\ell$, suitably rescaled.

## 4.1 Temporal context and interval timing

We consider a time interval estimation and reproduction task very similar to [24]. In each trial, the stimulus $s$ is a time interval (e.g., the interval between two flashes), drawn from a fixed experimental distribution, and the response $r$ is the reproduced duration (e.g., the interval between the second flash and a mouse click). Subjects perform in one or two conditions, corresponding to two different discrete uniform distributions of durations, either on a Short (494-847 ms) or a Long (847-1200 ms) range. Subjects are trained separately on each condition till they (roughly) learn the underlying distribution, at which point their performance is measured in a test session; here we only simulate the test sessions. We assume that the experimenter's goal is to faithfully recover the observer's priors, and we analyze the effect of different experimental designs on the reconstruction error.

To cast the problem within our framework, we need first to define the reference observer $\boldsymbol{\theta}^*$. We make the following assumptions: (a) the observer's priors (or prior, in only one condition) are smoothed versions of the experimental uniform distributions; (b) the sensory noise is affected by the scalar property of interval timing, so that the sensory mapping is logarithmic ($s_0 \approx 0$, $d = 1$); (c) we take average sensorimotor noise parameters from [24]: $\sigma = 0.10$, $\gamma = 0$, $\kappa = 0$, and $\rho_0 \approx 0$, $\rho_1 = 0.07$; (d) for simplicity, the internal likelihood coincides with the measurement distribution; (e) the loss function in internal measurement space is almost-quadratic, with $\sigma_\ell = 0.5$, $\gamma_\ell = 0$, $\kappa_\ell = 0$; (f) we assume a small lapse probability $\lambda = 0.03$; (g) in case the observer performs in two conditions, all observer's parameters are shared across conditions (except for the priors). For the inferred observer $\boldsymbol{\theta}$ we allow all model parameters to change freely, keeping only assumptions (d) and (g). We compare the following variations of the experimental setup:

1. BSL: The baseline version of the experiment, the observer performs in both the Short and Long conditions ($N_{\text{tr}} = 500$ each);

2. SRT or LNG: The observer performs more trials ($N_{\text{tr}} = 1000$), but only either in the Short (SRT) or in the Long (LNG) condition;

3. MAP: As BSL, but we assume a difference in the performance feedback of the task such that the reference observer adopts a narrower loss function, closer to MAP ($\sigma_\ell = 0.1$);

4. MTR: As BSL, but the observer's motor noise parameters $\rho_0, \rho_1$ are assumed to be known (e.g. measured in a separate experiment), and therefore fixed during the inference.

We sample from the approximate posterior density (Eq. 13), obtaining a set of sampled priors for each distinct experimental setup (see Supplementary Material for details). Figure 2 a shows the reconstructed priors and their central moments for the Short condition (results are analogous for the Long condition; see Supplementary Material). We summarize the reconstruction error of the recovered priors in terms of symmetric KL-divergence from the reference prior (Figure 2 b). Our analysis suggests that the baseline setup BSL does a relatively poor job at inferring the observers' priors. Mean and skewness of the inferred prior are generally acceptable, but for example the SD tends to be considerably lower than the true value. Examining the posterior density across various dimensions, we find that this mismatch emerges from a partial non-identifiability of the sensory noise, $\sigma$, and the motor noise, $w_1$ (Figure 2 c).[1] Limiting the task to a single condition with double number of trials (SRT) only slightly improves the quality of the inference. Surprisingly, we find that a design that encourages the observer to adopt a loss function closer to MAP considerably worsens the quality of the reconstruction in our model. In fact, the loss width parameter $\sigma_\ell$ is only weakly identifiable (Figure 2 d), with severe consequences for the recovery of the priors in the MAP case. Finally, we find that if we can independently measure the motor parameters of the observer (MTR), the degeneracy is mostly removed and the priors can be recovered quite reliably.

Our analysis suggests that the reconstruction of internal representations in interval timing requires strong experimental constraints and validations [12]. This worked example also shows how our framework can be used to rank experimental designs by the quality of the inferred features of interest (here, the recovered priors), and to identify parameters that may critically affect the inference. Some findings align with our intuitions (e.g., measuring the motor parameters) but others may be non-obvious, such as the bad impact that a narrow loss function may have on the inferred priors within our model. Incidentally, the low identifiability of $\sigma_\ell$ that we found in this task suggests that claims about the loss function adopted by observers in interval timing (see [24]), without independent validation, might deserve additional investigation. Finally, note that the analysis we performed is theoretical, as the effects of each experimental design are formulated in terms of changes in the parameters of the ideal reference observer. Nevertheless, the framework allows to test the robustness of our conclusions as we modify our assumptions about the reference observer.

## 4.2 Slow-speed prior in speed perception

As a further demonstration, we use our framework to re-examine a well-known finding in visual speed perception, that observers have a heavy-tailed prior expectation for slow speeds [9, 29]. The original study uses a 2AFC paradigm [9], that we convert for our analysis into an equivalent estimation task (see e.g. [30]). In each trial, the stimulus magnitude $s$ is speed of motion (e.g., the speed of a moving dot in deg/s), and the response $r$ is the perceived speed (e.g., measured by interception timing). Subjects perform in two conditions, with different contrast levels of the stimulus, either High ($c_{\text{High}} = 0.5$) or Low ($c_{\text{Low}} = 0.075$), corresponding to different levels of estimation noise. Note that in a real speed estimation experiment subjects quickly develop a prior that depends on the experimental distribution of speeds [30] – but here we assume no learning of that kind in agreement with the underlying 2AFC task. Instead, we assume that observers use their 'natural' prior over speeds. Our goal is to probe the reliability of the inference of the slow-speed prior and of the noise characteristics of the reference observer (see [9]).

We define the reference observer $\boldsymbol{\theta}^*$ as follows: (a) the observer's prior is defined in task space by a parametric formula: $p_{\text{prior}}(s) = (s^2 + s_{\text{prior}}^2)^{-k_{\text{prior}}}$, with $s_{\text{prior}} = 1$ deg/s and $k_{\text{prior}} = 2.4$ [29]; (b) the sensory mapping has parameters $s_0 = 0.35$ deg/s, $d = 1$ [29]; (c) the amount of sensory noise depends on the contrast level, as per [9]: $\sigma_{\text{High}} = 0.2$, $\sigma_{\text{Low}} = 0.4$, and $\gamma = 0$, $\kappa = 0$; (d) the internal likelihood coincides with the measurement distribution; (e) the loss function in internal measurement space is almost-quadratic, with $\sigma_\ell = 0.5$, $\gamma_\ell = 0$, $\kappa_\ell = 0$; (f) we assume a consider-

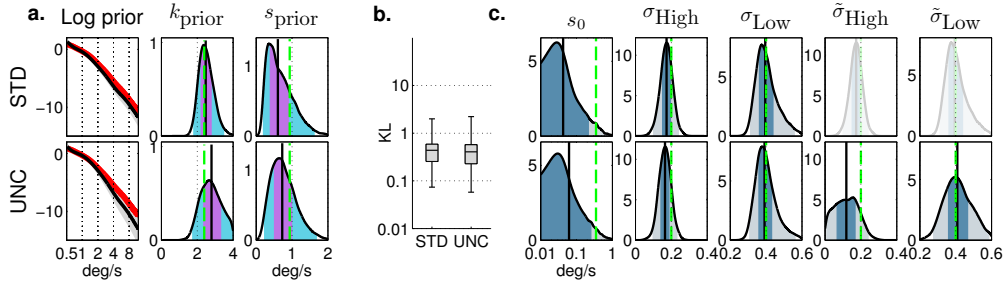

Figure 3: **Internal representations in speed perception.** Accuracy of the reconstructed internal representations (priors and likelihoods). Each row corresponds to different assumptions during the inference. **a:** The first column shows the reference log prior (thick red line) and the recovered mean log prior $\pm$ 1 SD (black line and shaded area). The other two columns display the approximate posteriors of $k_{\text{prior}}$ and $s_{\text{prior}}$, obtained by fitting the reconstructed 'non-parametric' priors with a parametric formula (see text). Each panel shows the median (black line), the interquartile range (dark-shaded area) and the 95 % interval (light-shaded area). The green dashed line marks the true value. **b:** Box plots of the symmetric KL-divergence between the reconstructed and reference prior. **c:** Approximate posterior distributions for sensory mapping and sensory noise parameters. In experimental design STD, the internal likelihood parameters ($\tilde{\sigma}_{\text{High}}, \tilde{\sigma}_{\text{Low}}$) are equal to their objective counterparts ($\sigma_{\text{High}}, \sigma_{\text{Low}}$).

able amount of reporting noise, with $\rho_0 = 0.3$ deg/s, $\rho_1 = 0.21$; (g) we assume a contrast-dependent lapse probability ($\lambda_{\text{High}} = 0.01$, $\lambda_{\text{Low}} = 0.05$); (h) all parameters that are not contrast-dependent are shared across the two conditions. For the inferred observer $\theta$ we allow all model parameters to change freely, keeping only assumptions (d) and (h). We consider the standard experimental setup described above (STD), and an 'uncoupled' variant (UNC) in which we do not take the usual assumption that the internal representation of the likelihoods is coupled to the experimental one (so, $\tilde{\sigma}_{\text{High}}, \tilde{\sigma}_{\text{Low}}, \tilde{\gamma}$ and $\tilde{\kappa}$ are free parameters). As a sanity check, we also consider an observer with a uniformly flat speed prior (FLA), to show that in this case the algorithm can correctly infer back the *absence* of a prior for slow speeds (see Supplementary Material).

Unlike the previous example, our analysis shows that here the reconstruction of both the prior and the characteristics of sensory noise is relatively reliable (Figure 3 and Supplementary Material), without major biases, even when we decouple the internal representation of the noise from its objective counterpart (except for underestimation of the noise lower bound $s_0$, and of the internal noise $\tilde{\sigma}_{\text{High}}$, Figure 3 c). In particular, in all cases the exponent $k_{\text{prior}}$ of the prior over speeds can be recovered with good accuracy. Our results provide theoretical validation, in addition to existing empirical support, for previous work that inferred internal representations in speed perception [9, 29].

## 5 Conclusions

We have proposed a framework for studying a priori identifiability of Bayesian models of perception. We have built a fairly general class of observer models and presented an efficient technique to explore their vast identifiability landscape. In one case study, a time interval estimation task, we have demonstrated how our framework could be used to rank candidate experimental designs depending on their ability to resolve the underlying degeneracy of parameters of interest. The obtained ranking is non-trivial: for example, it suggests that experimentally imposing a narrow loss function may be detrimental, under certain assumptions. In a second case study, we have shown instead that the inference of internal representations in speed perception, at least when cast as an estimation task in the presence of a slow-speed prior, is generally robust and in theory not prone to major degeneracies.

Several modifications can be implemented to increase the scope of the psychophysical tasks covered by the framework. For example, the observer model could include a generalization to arbitrary loss spaces (see Supplementary Material), the generative model could be extended to allow multiple cues (to analyze cue-integration studies), and a variant of the model could be developed for discrete-choice paradigms, such as 2AFC, whose identifiability properties are largely unknown.

## Footnotes

[1]This degeneracy is not surprising, as both sensory and motor noise of the reference observer $\boldsymbol{\theta}^*$ are approximately Gaussian in internal measurement space ($\sim$ log task space). This lack of identifiability also affects the prior since the relative weight between prior and likelihood needs to remain roughly the same.

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
