[Supplementary Material]

# A Framework for Testing Identifiability of Bayesian Models of Perception – Supplementary Material

**Luigi Acerbi**[1,2]     **Wei Ji Ma**[2]     **Sethu Vijayakumar**[1]

[1] School of Informatics, University of Edinburgh, UK
[2] Center for Neural Science & Department of Psychology, New York University, USA
{luigi.acerbi,weijima}@nyu.edu    sethu.vijayakumar@ed.ac.uk

## Contents

## 1   Introduction

Here we report supplementary information to the main paper, such as extended mathematical derivations and implementation details. For ease of reference, this document follows the same division in sections of the paper, and supplementary methods are reported in the same order as they are originally referenced in the main text.

## 2   Bayesian observer model

We describe here several technical details regarding the construction of the Bayesian observer model which are omitted in the paper.

## 2.1 Mapping to internal measurement space

The mapping to internal measurement space is a mathematical trick to deal with observer models whose sensory noise magnitude is stimulus-dependent in task space. For this derivation, let us assume that the measurement probability density, $p_{\text{meas}}(x|s)$, can be expressed as a Gaussian with stimulus-dependent noise:

$$p_{\text{meas}}(x|s) = \mathcal{N}\left(x|s, \sigma_{\text{meas}}^2(s)\right). \tag{S1}$$

In the case of Weber's law, we would have $\sigma_{\text{meas}}(s) = bs$, with $b > 0$ standing for Weber's constant (this feature of noise is called the *scalar property* in time perception [1, 2]).

The problem with Eq. S1 is that the measurement distribution is Gaussian but the likelihood (function of $s$) is not – which is unwieldy for the computation of the posterior. A solution consists in finding a transformed space in which the likelihood is (approximately) Gaussian. It is easy to show that a mapping of the form:

$$f(s) = \int_{-\infty}^{s} \frac{1}{\sigma_{\text{meas}}(s')} ds' \; + \; \text{const} \tag{S2}$$

achieves this goal. In fact, we can write an informal proof as follows:

$$\begin{aligned} f(x) &= f\left(s + \sigma_{\text{meas}}(s) \cdot \eta\right) \\ &\approx f(s) + f'(s) \cdot \sigma_{\text{meas}}(s) \cdot \eta \\ &= t + \eta \end{aligned} \tag{S3}$$

where $\eta$ is a normally distributed random variable with zero mean and unit variance. The second passage of Eq. S3 uses a first-order Taylor expansion, under the assumption that the noise magnitude is low compared to the magnitude of the stimulus. The last passage shows that the measurement variable is approximately Gaussian in internal measurement space with mean $t \equiv f(s)$ and unit variance.

For Weber's law, the solution of Eq. S2 has a logarithmic form $f(s) \propto \log s$, which is commonly used in the psychophysics literature. We want the mapping to cover both constant noise and scalar noise (and intermediate cases), so we consider a generalized transform, $f(s) \propto \log\left(1 + \frac{s}{s_0}\right)$, where the base magnitude parameter $s_0$ controls whether the mapping is purely logarithmic (for $s_0 \to 0$), linear (for $s_0 \to \infty$) or in-between [3]. For the paper, we further generalize the mapping, Eq. 2 in the main text, by adding a power exponent $d$ that allows to reproduce Steven's power law of sensation [4]. Note that the exponent $d$ has no effect if the mapping is (close to) purely logarithmic.

## 2.2 Moment-based parametrization of a unimodal mixture of two Gaussians

Let us consider a mixture of two Gaussian distributions:

$$p(s) = w\mathcal{N}\left(x|\mu_1, \sigma_1^2\right) + (1-w)\mathcal{N}\left(x|\mu_2, \sigma_2^2\right). \tag{S4}$$

We want to express its parameters ($w, \mu_i, \sigma_i$, for $i = 1, 2$) as a function of the standardized moments of the distribution: mean $\mu$, variance $\sigma^2$, skewness $\gamma$ and (excess) kurtosis $\kappa$, with the additional constraint of unimodality. The first two standardized moments are $\mu = 0$ and $\sigma^2 = 1$ (this is without loss of generality, as we may later rescale and shift the resulting distribution to match arbitrary values of $\mu$ and $\sigma^2$). Since there are five parameters and only four constraints, we will find a solution (or multiple solutions) as a function of the remaining parameter $w$.

- For the special case $\gamma = 0$ and $\kappa \leq 0$ we have:

$$\mu_1 = \sqrt[4]{-\frac{\kappa}{2}}, \qquad \mu_2 = -\mu_1, \qquad \sigma_1^2 = \sigma_2^2 = \sqrt{1 - \sqrt{-\frac{\kappa}{2}}}. \tag{S5}$$

- For the special case $\gamma = 0$ with $\kappa > 0$ the solutions are:

$$\mu_1 = \mu_2 = 0, \qquad \sigma_1^2 = 1 \mp \frac{\sqrt{(1-w)\kappa}}{\sqrt{3w}}, \qquad \sigma_2^2 = 1 \pm \frac{\sqrt{w\kappa}}{\sqrt{3(1-w)}}. \tag{S6}$$

- Finally, for the general case $\gamma \neq 0$:

$$\mu_1 = -\frac{1-w}{w} \cdot \mu_2$$

$$\mu_2 = \text{Roots}\left[\left(2 - 6w + 8w^2 - 6w^3 + 2w^4\right)y^6 + \left(4w^2\gamma - 12w^3\gamma + 8w^4\gamma\right)y^3 \right.$$
$$\left. + \left(3w^3\kappa - 3w^4\kappa\right)y^2 - w^4\gamma^2\right]$$

$$\sigma_1^2 = 1 + \frac{(w-1)\mu_2}{3w^4\gamma}\left[3w^3\kappa + (5-7w)w^2\gamma\mu_2 - 2(w-1)(1-w+w^2)\mu_2^4\right]$$

$$\sigma_2^2 = 1 + \frac{\kappa}{\gamma\mu_2} + \left(\frac{2}{3w} - \frac{7}{3}\right)\mu_2^2 - 2(w-1)(1+w^2-w)\frac{\mu_2^5}{3w^3\gamma} \tag{S7}$$

where Roots specifies the real roots of the polynomial in square brackets.

The final degree of freedom is chosen by picking the value of $w$ that locally maximizes the differential entropy of the distribution while respecting the requirement of unimodality and within the range $0.025 \leq w \leq 0.975$. The latter constraint is added to prevent highly degenerate solutions such as, e.g., $w \to 0$ with $\sigma_1^2 \to \infty$. At the implementation level, we do not perform these computations at every step but we precomputed a table that maps a pair of values $(\gamma, \kappa)$ to a parameter vector $w, \mu_i, \sigma_j$ for $j = 1, 2$ that uniquely identifies a mixture of two Gaussians.[1] This table also encodes the boundaries of $\gamma, \kappa$ since not all values are allowed (see Figure S1).

Figure S1: **Tabulated values of $w$ as a function of skewness and kurtosis.** The values of the mixing weight $w$ that respect the constraints of unimodality and $0.025 \leq w \leq 0.975$ cover a crescent-shaped region in the domain of skewness $\gamma$ and excess kurtosis $\kappa$ (shaded area). The black line represents the hard bound between skewness and kurtosis that applies to all univariate distributions, that is $\kappa \geq \gamma^2 - 2$.

## 2.3 Computation of the expected loss

The observer's prior is written as:

$$q_{\text{prior}}(t) = \sum_{m=1}^{M} w_m \mathcal{N}\left(t | \mu_m, a^2\right) \tag{S8}$$

where $a$ is the lattice spacing and we have defined $\mu_m \equiv \mu_{\min} + (m-1)a$ (see Eq. 4 in the paper). The internal measurement likelihood takes the form:

$$q_{\text{meas}}(x|t) = \sum_{j=1}^{2} \tilde{\pi}_j \mathcal{N}\left(x|t + \tilde{\mu}_j, \tilde{\sigma}_j^2\right) \tag{S9}$$

with $\tilde{\pi}_1 \equiv \tilde{\pi}$, $\tilde{\pi}_2 \equiv 1 - \tilde{\pi}$ (see the corresponding Eq. 3 in the paper). The posterior distribution is computed by multiplying Eq. S8 and S9:

$$
\begin{aligned}
q_{\text{post}}(t|x) &= \sum_{m=1}^{M} \sum_{j=1}^{2} w_m \tilde{\pi}_j \mathcal{N}\left(t|\mu_m, a^2\right) \mathcal{N}\left(t|x - \tilde{\mu}_j, \tilde{\sigma}_j^2\right) \\
&= \sum_{m=1}^{M} \sum_{j=1}^{2} \gamma_{mj} \mathcal{N}\left(t|\nu_{mj}, \tau_{mj}^2\right)
\end{aligned}
\tag{S10}
$$

obtained after some algebraic manipulations and where we have defined:

$$
\begin{aligned}
\gamma_{mj} &\equiv w_m \tilde{\pi}_j \mathcal{N}\left(\mu_m|x - \tilde{\mu}_j, a^2 + \tilde{\sigma}_j^2\right) \\
\nu_{mj} &\equiv \frac{\mu_m \tilde{\sigma}_j^2 + (x - \tilde{\mu}_j)a^2}{a^2 + \tilde{\sigma}_j^2} \\
\tau_{mj}^2 &\equiv \frac{a^2 \tilde{\sigma}_j^2}{a^2 + \tilde{\sigma}_j^2}.
\end{aligned}
\tag{S11}
$$

The loss function depends on the signed error in internal measurement space and is defined as:

$$\mathcal{L}\left(\hat{t} - t\right) = -\sum_{k=1}^{2} \pi_k^\ell \mathcal{N}\left(\hat{t} - t|\mu_k^\ell, {\sigma_k^\ell}^2\right) \tag{S12}$$

with $\pi_1^\ell \equiv \pi^\ell$ and $\pi_2^\ell \equiv 1 - \pi^\ell$ (see Eq. 6 in the paper). The expected loss for estimate $\hat{t}$, given measurement $x$, therefore takes the closed analytical form:

$$
\begin{aligned}
\mathbb{E}\left[\mathcal{L}; \hat{t}, x\right]_{q_{\text{post}}} &= \int q_{\text{post}}(t|x) \mathcal{L}\left(\hat{t} - t\right) dt \\
&= -\sum_{m=1}^{M} \sum_{j=1}^{2} \gamma_{mj} \sum_{k=1}^{2} \pi_k^\ell \int \mathcal{N}\left(t|\nu_{mj}, \tau_{mj}^2\right) \mathcal{N}\left(t|\hat{t} - \mu_k^\ell, {\sigma_k^\ell}^2\right) dt \\
&= -\sum_{m=1}^{M} \sum_{j=1}^{2} \gamma_{mj} \sum_{k=1}^{2} \pi_k^\ell \mathcal{N}\left(\hat{t}|\nu_{mj} + \mu_k^\ell, \tau_{mj}^2 + {\sigma_k^\ell}^2\right).
\end{aligned}
\tag{S13}
$$

Eq. S13 generalizes a previous result [5, Eq. 4] to the case of likelihoods and loss functions that are mixtures of Gaussians. Thanks to the expression of Eq. S13 as a mixture of Gaussians, the global minimum of the expected loss can be found very efficiently through an adaptation of Newton's method [5, 6]. Note that computational efficiency is not merely a desirable feature, but rather a key requirement for tractability of our analysis of the complex parameter space. In Section 5.1 we discuss how the framework can be generalized to a loss function whose error is computed in arbitrary spaces.

## 2.4 Mapping densities from task space to internal measurement space and vice versa

Variables are mapped from task space to internal measurement space (and vice versa) through the mappings described in Eq. 2 in the main text. Mapping of densities needs to take into account the Jacobian of the transformation.

A distribution $p(s)$ in task space is converted into internal measurment space as:

$$q(t) = \left|f^{-1'}(t)\right| p(f^{-1}(t)) = \left[\frac{s_0}{Ad}\left(e^{\frac{t-B}{A}} - 1\right)^{\frac{1}{d}-1} e^{\frac{t-B}{A}}\right] p(f^{-1}(t)). \tag{S14}$$

Conversely, the inverse transform of a density $q(t)$ from internal measurement space to task space is:

$$p(s) = |f'(s)| \, q(f(s)) = \left[ \frac{Ad \, (s/s_0)^{d-1}}{1 + (s/s_0)^d} \right] q(f(s)). \tag{S15}$$

## 3  Model identifiability

A pivotal role in our a priori identifiability analysis is taken by the equation that links the expected log likelihood to the KL-divergence between the response distributions. Here we show the derivation.

### 3.1  Derivation of Eq. 12 in the paper

We want to find a closed-form solution for Eq. 11 in the paper:

$$\langle \log \Pr(\mathcal{D}|\boldsymbol{\theta}) \rangle = \int_{|\mathcal{D}|=N_{\text{tr}}} \log \Pr(\mathcal{D}|\boldsymbol{\theta}) \Pr(\mathcal{D}|\boldsymbol{\theta}^*) \; d\mathcal{D}. \tag{S16}$$

First, we divide the dataset $\mathcal{D}$ as follows. Recall that the experiment presents a discrete set of stimuli $s_i$ with relative frequency $P_i$, for $1 \leq i \leq N_{\text{exp}}$. We assume that the number of trials for each stimulus is allocated a priori to match relative frequencies (a common practice in psychophysical experiments). Therefore, dataset $\mathcal{D}$ can be divided in $N_{\text{exp}}$ sub-datasets $\mathcal{D}_i$ with respectively $P_i N_{\text{tr}}$ trials each. Assuming independence between trials and thanks to linearity of the expectation operator, we can write:

$$\langle \log \Pr(\mathcal{D}|\boldsymbol{\theta}) \rangle = \sum_{i=1}^{N_{\text{exp}}} \langle \log \Pr(\mathcal{D}_i|\boldsymbol{\theta}) \rangle \tag{S17}$$

where each sub-dataset $\mathcal{D}_i$ only contains trials that show a specific stimulus $s_i$. In the following we compute the expectation of the log likelihood for a sub-dataset with a single stimulus.

Let us consider a sub-dataset $\mathcal{D}_i$ with $N \equiv P_i N_{\text{tr}}$ trials and stimulus $s_i$. The true distribution of responses in each trial is assumed to be stationary with distribution $p(r) \equiv p_{\text{resp}}(r|s_i, \boldsymbol{\theta}^*)$, whereas the distribution of responses according to model $\boldsymbol{\theta}$ is represented by $q(r) \equiv p_{\text{resp}}(r|s_i, \boldsymbol{\theta})$. The expected log likelihood of the sub-dataset for model $\boldsymbol{\theta}$ under true model $\boldsymbol{\theta}^*$ is:

$$
\begin{aligned}
\langle \log \Pr(\mathcal{D}_i|\boldsymbol{\theta}) \rangle_{\Pr(\mathcal{D}_i|\boldsymbol{\theta}^*)} &= \int \Pr(r_1, \ldots, r_N|\boldsymbol{\theta}^*) \log \Pr(r_1, \ldots, r_N|\boldsymbol{\theta}) \, dr_1 \times \ldots \times dr_N \\
&= \int \Pr(r_1, \ldots, r_N|\boldsymbol{\theta}^*) \left[ \log \prod_{j=1}^{N} q(r_j) \right] dr_1 \times \ldots \times dr_N \\
&= \sum_{j=1}^{N} \int \Pr(r_1, \ldots, r_N|\boldsymbol{\theta}^*) \log q(r_j) \, dr_1 \times \ldots \times dr_N \\
&= N \int p(r) \log q(r) \, dr \\
&= - P_i N_{\text{tr}} \cdot [D_{\text{KL}}(p||q) + H(p)]
\end{aligned}
\tag{S18}
$$

where $D_{\text{KL}}(p||q)$ is the Kullback-Leibler (KL) divergence, a non-symmetric measure of the difference between two probability distributions widely used in information theory, and $H(p)$ is the (differential) entropy of $p$. The last passage follows from the definition of cross-entropy [7]. Note that the entropy of $p$ does not depend on $\boldsymbol{\theta}$, so the entropy term is constant for our purposes. Combining Eqs. S17 and S18 we obtain Eq. 12 in the paper.

## 4  Supplementary methods and results

We report here additional details and results omitted for clarity from the main text.

### 4.1 Sampling from the approximate expected posterior density

The observer models we consider in the paper have 26-41 parameters, which correspond to a fairly high-dimensional parameter space. We assumed indepedent, non-informative priors on each model parameter, uniform on a reasonably inclusive range. Some parameters that naturally cover several orders of magnitude (e.g., the mixing weights $w_m$, for $1 \leq m \leq M$) were represented in log scale.[2] Also, the kurtosis parameters of likelihoods and loss function ($\kappa$, $\tilde{\kappa}$, $\kappa_\ell$) were represented in a transformed kurtosis space with $\kappa' \equiv \sqrt{\kappa + 2}$ (in this space, skewness and kurtosis are on a similar scale; the hard bound $\kappa \geq \gamma^2 - 2$ becomes $\kappa' \geq |\gamma|$).

We explored a priori identifiability in the large parameter space by sampling observers from the approximate expected posterior density, Eqs. 12 and 13 in the paper, via an adaptive MCMC algorithm [9]. Note that we computed the KL-divergence between the (rescaled) response distributions of a candidate model $\boldsymbol{\theta}$ and of the reference model $\boldsymbol{\theta}^*$, Eq. 12, only inside the range of experimental stimuli (this is equivalent to the experimental practice of discarding responses outside a certain range, to avoid edge effects). For each specific experimental design, we ran 6-10 parallel chains with different starting points near $\boldsymbol{\theta}^*$ ($5 \cdot 10^3$ burn-in steps, $5 \cdot 10^4$ to $2 \cdot 10^5$ samples per chain, depending on model complexity). To check for convergence, we computed Gelman and Rubin's potential scale reduction statistic $R$ for all parameters [10]. Large values of $R$ denote convergence problems whereas values close to 1 suggest convergence. For all experimental designs and parameters, $R$ was generally $\lesssim 1.1$. Paired with a visual check of the marginal pdfs of the sampled chains, this result suggests a resonable degree of convergence. Finally, chains were thinned to reduce autocorrelations, storing about $N_{\text{smpl}} = 10^4$ sampled observers per experimental design.

As an additional consistency check, we performed a 'posterior predictive check' (see e.g. [11]) with the sampled observers, that is we verified that the predicted behavior of the sampled observers matches the behavior of the reference observer across some relevant statistics (if not, it means that the sampling algorithm is not working correctly). We chose as relevant summary statistics the means and standard deviations of the observers' responses, as a function of stimulus $s_i$ and experimental condition (computed for each sampled observer via Eq. 9 in the paper). We found that the predicted response means were generally in excellent agreement with the 'true' response means of the reference observer. Distributions of predicted response variances showed some minor bias, but were still in good statistical agreement with the true response variance.

### 4.2 Temporal context and interval timing

The set of stimuli $s_i$ used in the experiment is comprised of $N_{\text{exp}} = 11$ equiprobable, regularly spaced intervals over the relevant range (Short 494-847 ms, Long 847-1200 ms) [2].

To reconstruct the observer's average prior (Figure 2 a in the paper), for each sampled observer, we computed the prior in internal space (Eq. 4 in the paper) and transformed it back to task space via Eq. S15; the mean prior is obtained by averaging all sampled priors. We also computed the first four central moments of each sampled prior in task space, whose distributions are shown in Figure 2 a in the main text. The reconstruction error for each sampled prior was assessed through the symmetric KL-divergence with the prior of the reference observer (the standard KL-divergence produces similar results).

Note that observer models BSL, MAP and MTR were tested on both the Short and Long ranges (models SRT and LNG were simulated only on either the Short or the Long range). Figure 2 in the paper reports only data for the Short range; we show here the priors recovered in the Long range condition for the same models (Figure S2). Results are qualitatively similar to what we observed for the Short range, with similar deviations from the reference prior and the same ranking between experimental designs.

### 4.3 Slow-speed prior in speed perception

The set of stimuli $s_i$ is comprised of $N_{\text{exp}} = 6$ equiprobable motion speeds: $s \in \{0.5, 1, 2, 4, 8, 12\}$ deg/s [3].

Figure S2: **Internal representations in interval timing (Long condition).** Accuracy of the reconstructed priors in the Long range; each row corresponds to a different experimental design. Figure 2 in the main text reports data for the Short range in the same format. See caption of Figure 2 in the main text for a detailed legend. **a:** The first column shows the reference prior and the recovered mean prior. The other columns display the recovered central moments of the prior. **b:** Box plots of the symmetric KL-divergence between the reconstructed priors and the prior of the reference observer.

We reconstructed the observer's average *log* prior in task space (Figure 3 a in the paper) for each sampled observer. To capture the shape of the sampled priors, we fit each of them with a parametric formula: $\log q(s) = -k_{\text{prior}} \log \left( s^2 + s^2_{\text{prior}} \right) + c_{\text{prior}}$, via least-squares estimation. The distribution of fitted parameters $k_{\text{prior}}$ and $s_{\text{prior}}$ is shown in Figure 3 a in the main text.

We show here the results for an additional observer model (FLT) which incorporates a uniformly flat prior (Figure S3). The model inference correctly recovers a flat prior with exponent $k_{\text{prior}} \approx 0$ (compare it with Figure 3 in the paper).

Figure S3: **Internal representations in speed perception (flat prior).** Accuracy of the reconstructed internal representations (priors and likelihoods) for an observer with a uniformly flat prior. Figure 3 in the main text reports data for other two observer models in the same format. See caption of Figure 3 in the main text for a detailed legend. **a:** The first panel shows the reference log prior and the recovered mean log prior. The other two panels display the approximate posteriors of $k_{\text{prior}}$ and $s_{\text{prior}}$. **b:** Box plot of the symmetric KL-divergence between the reconstructed and reference prior. **c:** Approximate posterior distributions for sensory mapping and sensory noise parameters.

# 5   Extensions of the observer model

We discuss here an extension of the framework presented in the main text.

## 5.1 Expected loss in arbitrary spaces

In the paper we have used a loss function that depends on the error, i.e. on the difference between the estimate and the true value of the stimulus, in internal measurement space (Eq. S12). However, we might want to compute the error in task space, or more in general in an arbitrary loss space defined by a mapping $g(s) : s \rightarrow l$ parametrized by $s_0^\ell$ and $d^\ell$ (see Eq. 2 in the paper). Ideally, we still want to find a closed-form expression for the expected loss. We can write the loss function in the new loss space as:

$$\mathcal{L}\left(g\left(\hat{s}\right) - g\left(s\right)\right) = \mathcal{L}\left(g\left(f^{-1}(\hat{t})\right) - g\left(f^{-1}(t)\right)\right) = \mathcal{L}\left(h(\hat{t}) - h(t)\right) \tag{S19}$$

where we have defined the composite function $h \equiv g \circ f^{-1}$. Clearly the original expression of the loss in internal measurement space is recovered if $g \equiv f$. We can rewrite the expected loss as follows:

$$\mathbb{E}\left[\mathcal{L}\right]_{q_{\text{post}}}(\hat{t}) = -\sum_{m=1}^{M}\sum_{j=1}^{2}\gamma_{mj}\sum_{k=1}^{2}\pi_k^\ell \mathcal{N}\left(h(\hat{t})|\nu_{mj} + \mu_k^\ell, \tau_{mj}^2 + \sigma_k^{\ell\,2}\right) \\ \times \int \mathcal{N}\left(h(t)|\bar{\nu}_{mjk}(\hat{t}), \bar{\tau}_{mjk}^2\right) dt, \tag{S20}$$

where we have defined:

$$\bar{\nu}_{mjk}(\hat{t}) \equiv \frac{\nu_{mj}\sigma_k^{\ell\,2} + \left(h(\hat{t}) - \mu_k^\ell\right)\tau_{mj}^2}{\tau_{mj}^2 + \sigma_k^{\ell\,2}}, \qquad \bar{\tau}_{mjk} \equiv \frac{\tau_{mj}^2 \sigma_k^{\ell\,2}}{\tau_{mj}^2 + \sigma_k^{\ell\,2}}. \tag{S21}$$

In order to perform the integration in Eq. S20, we Taylor-expand $h(t)$ up to the first order around the mean of each integrated Gaussian, $\bar{\nu}_{mjk}(\hat{t})$. We can perform this linearization without major loss of accuracy since the Gaussians in the integral are narrow, their variance bounded from above by $a^2$ ($\bar{\tau}_{mjk}^2 < \tau_{mj}^2 < a^2$, see Eqs. S11 and S21). The integration yields:

$$\mathbb{E}\left[\mathcal{L}\right]_{q_{post}}(\hat{t}) \approx -\sum_{m=1}^{M}\sum_{j=1}^{2}\gamma_{mj}\sum_{k=1}^{2}\pi_k^\ell \mathcal{N}\left(h(\hat{t})|\nu_{mj} + \mu_k^\ell, \tau_{mj}^2 + \sigma_k^{\ell\,2}\right)\frac{1}{h'\left(\bar{\nu}_{mjk}(\hat{t})\right)}. \tag{S22}$$

Eq. S22 is not a regular mixture of Gaussians, but we can write its first and second derivative analytically, which in principle allows to apply Newton's method for numerical minimization. This derivation shows that the techniques developed in the paper can be extended to the general case of a loss function based in an arbitrary space (including, in particular, task space).

## Acknowledgments

We thank Jonathan Pillow, Paolo Puggioni, Peggy Seriès, and three anonymous reviewers for useful comments on earlier versions of the work.

## Footnotes

[1]Thanks to symmetries we need to precompute the table only for $\gamma \geq 0$ and we flip the sign of $\mu_1$ and $\mu_2$ for $\gamma < 0$.

[2]Note that a uniform prior in log space implies a prior of the form $\sim 1/x$ in standard space [8].