[Reviews · NeurIPS 2014]

Submitted by Assigned_Reviewer_18

In this article, the authors propose a framework for performing model comparison of Bayesian models on behavioral data. To do so, they summarize the Bayesian Decision Theory framework, pinpoint areas of non-identifiability, and outline the types of constraints that can be used to make each term in the Bayesian framework identifiable. They then make assumptions to constrain each term in the Bayesian framework, explore how differentiable parameter values are in their model, and apply the technique to two studies that use Bayesian decision theory to explain behavioral responses: time interval estimation and motion perception.

Issues of identifiability of internal representations and processes have been prominent issues within cognitive science and psychology for decades. For example, Anderson (1978) analyzed when representational formats can mimic each other, proving that they are indistinguishable without some constraints on processes (whenever the representations make equivalent distinctions of their inputs).
In recent years, it is one of the major drawbacks of the Bayesian framework that has hindered its widespread adoption (e.g., Jones & Love, 2011; Bowers & Davis, 2012 – the latter directly addresses speed perception as an example).

Although the article touches on an issue of great theoretical importance, is well written, and generally does so in a reasonable and effective manner, I have two major reservations with it. First, it ignores the large previous literature on model comparison within cognitive science and psychology, which makes it difficult to evaluate its contribution appropriately. The general framework for model comparison proposed by the authors is not so different from some of this previous work (e.g., Pitt, Myung, & Zhang, 2002; Navarro, Pitt, & Myung, 2004). Second, the authors make assumptions that seem arbitrary at times and it is unclear how much of their framework depends on them. For example, why should $p_{meas}$ be defined by equation 3? Is defining the values of the parameters in Equation 12 really enough to define and capture all the characteristics of a cognitive or perceptual process? I understand that it is robust and can express a large class of sensory models. However, it is not clear that is a sufficient justification for the assumption and at the very least, the article should discuss and justify the assumptions used to constrain the Bayesian models so that they are identifiable.

Minor comments:
Line 53: “It is a trivial mathematical fact…” It might be more helpful to explain why it is true rather than declare it trivial. It might not be trivial for all readers.

Line 73: I believe the second “nor” should be an “or”.

Figure 2: What do the star and dashed lines denote? From the caption, it sounds like they both denote the true value.

References

Anderson, J. R. (1978). Arguments concerning representations for mental imagery. Psychological Review, 85, 249-277.

Bowers, J. S., & Davis, C. J. (2012). Bayesian Just-So Stories in Psychology and Neuroscience. Psychological Bulletin, 138(3), 389-414.

Jones, M., & Love, B. C. (2011). Bayesian fundamentalism or enlightenment? On the explanatory status and theoretical contributions of Bayesian models of cognition. Behavioral and Brain Sciences, 34, 169-188.

Navarro, D. J., Pitt, M. A., & Myung, I. J. (2004). Assessing the distinguishability of models and the informativeness of data. Cognitive Psychology, 49, 47-84.

Pitt, M. A., Myung, I. J., & Zhang, S. (2002). Toward a Method of Selecting Among Computational Models of Cognition. Psychological Review, 109(3), 472-491. (also see Myung, Pitt, Zhang, & Balasubramanian, 2000 NIPS 13).

Summary: A well-written article on an issue of great importance, but missing links to previous related work.

Submitted by Assigned_Reviewer_24

The manuscript tackles the important issue in cognitive modeling of
model identifiability. The manuscript is nicely written and quite clear.
However, the work is premised on a long list of assumptions that one
must buy into and make sense of, and this list is quite long relative
to the impact of the key results presented. Because the methodology
is interesting and innovative, I judge the work to be above threshold
for presentation at NIPS.

Insofar as key results go: In section 4.1, the interchangeability of sensory
and motor noise was little surprise (I questioned incorporating both into
the model as I was reading). It would help to indicate which condition
differences were reliable. (Maybe I'm just too dumb to read box plots.)
In section 4.2, isn't it essential to show that if the reference
observe did not have a slow-speed prior, the model could recover this
fact as well?

Specific comments:

line 106: Why is the stimulus drawn from a discrete distribution when it in
fact a continuous variable? I assume this is due to the typical
design of experiments, where the stimulus level is the independent
variables. But the authors may wish to clarify this point.

line 134: As the authors note, the log-stimulus transform (Equation 2) is
motivated by psychophysical laws. It seems that the success of their work
may be due in large part to the incorporation of this transform.
That is, the transform imposes a fairly strong constraint on inference
and thus, this work is really saying that Bayesian models are identifiable
given strong assumptions about representational transformations in the mind.

line 143: Do the constraints in Eq 4 and the maximization of differential
entropy ensure that Eqn 3 is fit to be a unimodal distribution? It seems like
the fit could yield in principle a bimodal distribution.

line 158: Why should the observer's prior favor the lattice representation
over a uniform representation? What does the user know about the true
stimulus distribution that allows them to have a prior which is 50% wider
than the true range?

Figure 1: Even after staring at the figure, I feel more confused by it
than enlightened. It seems like it would be clearer if Figure 1b were
integrated into Figure 1a, and p_est(s^*|x) were removed from the
Figure, since it is a derived inference.

line 252: Can the authors comment on the conditions on N that
make Equation 14 approximately correct, i.e., when Stirling's approximation
of the factorial and the holds, and under what circumstances E[xlogx] ~=
E[x]logE[x]. I have to say that by the point where the authors come up
with Equation 15---having abandoned priors on \theta and made some
questionable approximations---I feel that Equation 15 is justified only
by the fact that it yields an intuitive rule for assessing similarity
of predictions.

line 262, "sample from this unnormalized density": Which density? The dataset
distribution?

line 365: Why does learning of priors not matter in 2AFC?
Summary: Innovative approach, important problem, less than overwhelming results

Submitted by Assigned_Reviewer_33

The paper describes a Bayesian model of perception (mathematical equations for sensory noise, likelihood function, prior probability distribution, loss function, response distribution). A significant strength of this model is that it combines both robustness and mathematical convenience. That is, the model can handle many different types of situations (e.g., stimulus-dependent noise) and still be computationally convenient (e.g., inference is tractable). The paper places emphasis on the fact that the model is identifiable (i.e., two parameter estimates can be easily evaluated and distinguished).

I like this paper and would like to see it accepted to the NIPS conference. The proposed model provides a mathematically elegant framework for formalising perceptual estimation in many experiments. At the same time, I have some reservations. I believe that the proposed model is very much in the same spirit and style as other models that have recently appeared in the literature (often in the journal PLoS Computational Biology), and thus the novel contribution of this paper is incremental. For example, the reader may consider the following papers:

Acerbi, L., Wolpert, D. M., & Vijayakumar, S. (2012). Internal representations of temporal statistics and feedback calibrate motor-sensory interval timing. PLoS Computational Biology, 8 (11), e1002771.

Acerbi, L., Vijayakumar, S., & Wolpert, D. M. (2014). On the origins of suboptimality in human probabilistic inference. PLoS Computational Biology, 10 (6), e1003661.

The current submission is not identical to these (or other) earlier articles but it is similar enough (to these and other earlier articles) that I think that it is fair to state that the contribution of the current submission is "incremental".

I suggest that the authors revise the manuscript by including a new section in which they review recently proposed Bayesian models of perception, and clearly state why the model proposed in the current submission is a significant advance over previously published models.

One more comment. The submission emphasizes the fact that the model is mathematically identifiable. That is true, but parameters are only identifiable due to the mathematical assumptions of the model. The term "identifiable" can be used in other ways. For example, the model could not be used to firmly distinguish whether an experimental subject's errors are due to sensory noise, decision noise, or response noise. As a result, it would be justified to say that the model does not help with the identifiability issue with which most perceptual scientists are concerned. Please comment on this in the revised manuscript.
Summary: Mathematically elegant framework for formalising perceptual estimation. However, the novel contribution is incremental (closely related work already exists in the literature).
Author Feedback
Author rebuttal: We thank all anonymous reviewers, whom we refer to as R1, R2 and R3, for their detailed feedback. Many comments represent minor suggested changes, such as additional references and clarifications, that we will incorporate in the revised manuscript; we respond here to the major points.

Mainly, we would like to clarify the scope of our paper and in what aspects it builds upon existing literature on Bayesian modelling of perception and model identifiability.

We have proposed a novel, unified observer model family that is able to capture in a single continuous parametrization a large number of existing models/tasks of sensorimotor estimation (see Section 2, lines 87-96). Our work extends previous models that have been applied to specific tasks (e.g., references [9-10] in the paper and those cited by R3) and is, to our knowledge, the first explicit attempt at unifying a whole class of perceptual models and tasks in a way which is both complete and computationally efficient.
In response to some concerns of R1, we would like to point out that the explicit scope of the proposed model class is precisely Bayesian models of perception (BMP) and sensorimotor estimation as defined in Section 2, and not more general (Bayesian) models used in psychology and cognitive science. Restriction to perception allows us to adopt some specific assumptions, such as that the stimulus-dependence of sensory variability can be described by a choice of parameters of the generalized psychophysical law (Eq. 2) and that the sensory noise is unimodally distributed around the stimulus in internal measurement space, although not necessarily Gaussian (Eq. 3). The formulated assumptions cover the majority of studies in the field of BMP, although we agree that there are several ways in which additional features could be included in the framework to extend its range of applicability (see Conclusions and Section 5 of the Supplement).

We thank R1 for pointing out that our approach is similar in spirit to previous work on model identifiability in psychology (e.g., Navarro et al. 2004) which we will refer to in the revised manuscript. Our paper however differs in several major aspects. Here we introduced a framework that specifically allows to explore the inner workings of BMP and how simultaneous changes to priors, likelihoods and loss functions affect psychophysical performance. Flexibility of the model (such as the “nonparametric” prior, Eq. 5) implies a large number of parameters (N = 26-41 in our analyses).
Novel elements of our proposal that make this problem tractable are both the definition of the continuous-parameter model (Section 2) and a (relatively) fast approximate technique to explore the identifiability landscape (Section 3). Exploring the identifiability landscape via resampling, as usually proposed (e.g., Navarro et al. 2004), would be computationally intractable. Our framework as a whole is, therefore, narrower in scope than generic techniques for model identifiability, but allows to obtain answers within the domain of interest (BMP) that would be out of reach with traditional methods.

Also, just to clarify a potential source of confusion, we did not build our model class specifically in order to make it identifiable. We built our model class so that it is computationally efficient and able to express a vast number of specific tasks and observer models in the literature. Incidentally, several of the typical assumptions in psychophysics may help make the final model identifiable, as we remark in lines 63-64 (and also noted by R2).

Our model class can then be used as a tool to verify, given a specific reference model (and experimental layout), what is the theoretical identifiability landscape in that region of model space. We presented two examples (Section 4) in which the variables of interest were the shapes of the prior and some characteristics of the noise or the loss function, but the experimenter may choose to focus on other aspects. For example, he or she may look for an experimental design that improves information (better identifiability) along one dimension at the expense of discriminability along other dimensions. To respond to the last remark of R3, in theory the framework could be used to look for experimental designs that allow to better distinguish between different sources of noise (if such designs exist).

We conclude by responding to some specific doubts of R2:
- We included both sensory and motor noise in the model in Section 4.1 since we are reproducing the observer model in reference [23].
- For Section 4.2, we also have data showing that we can recover the correct prior and parameters of a reference observer with a flat prior; we will include this.
- line 106: The experimental distribution of stimuli is often discrete in these studies (or can be appropriately binned with minor loss of generality). Discretization is necessary for Eq. 14.
- line 143: We do assume unimodality (see Section 2.2 in the Supplement).
- line 158: We are using a mixture of Gaussians on a lattice as a means to approximate any continuous distribution in a “nonparametric” way (i.e. with potentially many parameters). Prior expectations (e.g., slow-speed prior) and experimental distributions (e.g., test stimuli used in the task) may be quite different, so we allow for this difference (see Section 4).
- line 252: Stirling’s approximation is very efficient already for n > 10. Eq. 15 is motivated by Eq. 14 but we agree that it includes several approximations.
- line 262: We sample from Eq. 15, defined on line 261.
- line 365: 2AFC usually does not provide feedback and conveys little information, so learning/update of priors is assumed to be very slow. Otherwise studies of ecological priors (e.g., [10-11]) would be impossible.

We thank again all reviewers for their useful suggestions, and we hope that this reply will clarify doubts and address all the concerns raised.